# The effect of anchors and social information on behaviour

Tanya O'Garra [1,2☯]*, Matthew R. Sisco[2☯]

**1** Department of Economics, Middlesex University, London, United Kingdom, **2** Center for Research on Environmental Decisions, Columbia University, New York, New York, United States of America

☯ These authors contributed equally to this work.
* t.ogarra@mdx.ac.uk

**Data Availability Statement:** All relevant data is within the paper and its Supporting Information files.

**Funding:** The author(s) received no specific funding for this work. It was funded through the first author's personal research funds by the Earth

## Abstract

We use a 'multi-player dictator game' (MDG), with 'social information' about the monetary transfer made by a previous dictator to a recipient, to examine whether average contributions as well as the behavioural strategy adopted are affected by the first amount presented (the 'anchor') using a sequential strategy elicitation method. We find that average contributions are positively affected by the anchor. The anchor is also found to influence the behavioural strategy that individuals adopt, such that low anchors significantly increase the likelihood that players will adopt unconditional self-interested strategies, whereas high anchors increase the likelihood of adopting giving strategies. The distribution of strategies–and hence, the distribution of behavioural 'types'—is therefore affected by the initial conditions of play, lending support to the notion that behavioural strategies are context dependent.

## Introduction

This paper reports results of an experiment that examines the impact of an initial piece of information—or *'anchor'*—on redistribution choices in response to social information.

Anchoring is a well-established cognitive phenomenon describing the tendency of individuals to make judgments that are biased towards the first piece of information they receive [1,2]. Most anchoring studies examine the impact of anchors on numerical judgments (e.g. [1]), beliefs (e.g. [3,4]), and elicited preferences (e.g. [5]). Anchors are also found to affect actual behaviour, including consumer bidding in auctions (e.g. [6, 7, 8]), consumer purchases [9], and valuations of consumer goods with probabilistically binding choices (e.g. [10, 11]) although Fudenberg et al (2012) [12] do not replicate the findings in [11].

There has been much less research on the effects of anchors on pro-social behaviour, such as cooperation and redistribution, and what little evidence there is, is mixed. For example, Cappelletti, Güth, & Ploner, M. (2011) [13] and Luccasen (2012) [14] both fail to find evidence of anchoring effects on cooperation behaviour using public goods games, whereas Fosgaard & Piovesan (2015) [15] find that subjects playing a public goods game with default options (using the strategy method) anchor their subsequent decisions to the default. The evidence is similarly inconclusive with respect to anchoring effects on redistribution behaviour. Raihani &

Institute Postdoctoral Fellowship programme at
Columbia University.

**Competing interests:** The authors have declared
that no competing interests exist.

McAuliffe (2014) [16] find that numerical anchors (based on player's ages) have no effect on monetary transfers in a dictator game–although we note that the treatments analysed for anchoring in [16] were intended as control treatments and were not designed to elicit an anchoring effect. On the other hand, Dhingra et al (2012) [17] find evidence that choices in a dictator game with default options are anchored to the defaults; they term this a "default pull" although it corresponds essentially to an anchoring effect.

We aim to add to this limited literature, by asking the following questions: firstly, can an initial piece of information alter the *amount* that an individual redistributes? By 'redistribution', we refer to decisions to share wealth with others, with no expectation or possibility of benefitting materially from redistribution. Secondly, if anchors do affect the amount that individuals redistribute, might they also affect how the individual perceives the situation and hence, their *behavioural strategy*?

The behavioural strategies that people adopt in economic experiments are often used to classify people into social 'types', such as 'conditional co-operators' or 'free riders' (e.g. [18, 19]). The general understanding is that these different behavioural strategies reflect underlying social preferences, such as 'altruism', 'reciprocity' or 'warm glow'. For example, redistribution behaviour in economic experiments is often considered indicative of altruistic preferences [20], while contributions to the public good are considered to reflect reciprocity or conformity [21].

However, a growing number of studies are finding that the specific behavioural strategies that individuals adopt—and hence the distribution of 'types'–are susceptible to contextual factors, such as the frame (e.g. [22]), and how choices are elicited (e.g. [23,15]). For example, Dariel (2018) [23] find that changing the way in which conditional strategies are elicited in a public goods game radically changes the proportions of conditional co-operators and free riders. This suggests that the behavioural strategies that individuals adopt may be context-dependent [24,25].

To address this question, we examine the behavioural strategies that individuals adopt *in response to social information*. The effect of social information on redistribution decisions has been extensively explored (e.g. [26–28]), and the general finding is that on average, people positively condition the amounts they give to the amounts given by others. However, there is heterogeneity in how individuals respond to social information, with some people positively conditioning their choices to those of others, some negatively conditioning their choices and others unaffected [29,30]. We ask whether the distribution of behavioural types in this context is sensitive to anchoring effects. This is exploratory research, and as such, we have no expectations about the size or direction of anchoring effects on the distribution of 'types' in the population under study. Our aim is mainly to identify whether the choice of behavioural strategy is affected by normatively irrelevant contextual factors, such as anchors.

To this end, we use a 'multi-player dictator game' (MDG), in which there is a first mover (FM) who makes an initial visible monetary transfer to recipients in the group, and second movers (SM) who make transfer choices in response to all possible FM choices using a sequential strategy method. The strategy method involves players providing contingent responses to a range of possible actions by a peer. Individual 'types' are classified based on the full vector of responses to FM transfers, as either: 'conformists' (positive relationship), 'compensators' (negative relationship), 'self-interested' (fixed zero transfer) or 'unconditional givers' (fixed positive transfer) types. The impact of anchors on the distribution of types is ascertained by randomly presenting different SMs with different starting values in the sequential strategy elicitation exercise and examining whether this initial amount affects the distribution of SM types. To the best of our knowledge, this is the first study to examine anchoring effects using the sequential strategy method.

Overall, we show evidence of an anchoring effect, with average transfers influenced by the initial amount that SMs must respond to using the sequential strategy method. We also find

that anchors affect the distribution of 'types', such that the likelihood of choosing an unconditional self-interested strategy is greater in response to low value anchors than high value anchors. This suggests that the adoption of self-interested strategies may be at least partly determined by contextual factors, such as anchors.

We consider this to be an important investigation for various reasons. Firstly, individuals are regularly faced with new redistribution decisions, for example, in the form of charitable appeals. If the initial piece of information determines the entire strategies adopted by potential donors, then it suggests that initial information has an inordinate influence on all the related decisions that follow. The practical value of this finding is highly significant, as anchoring effects could potentially be harnessed not only to 'nudge' individuals towards single instances of fair sharing, but towards the adoption of more persistent redistributive behaviour. Additionally, from a theoretical perspective, the behavioural strategy that an individual adopts is expected to reflect preferences. Assuming preferences to be stable and well-defined, if anchors cause a change in the distribution of behavioural strategies, this may suggest that such strategies (such as 'self-interest' or 'conformity') are not fixed and may actually reflect different psychological processes and motivations interacting with contextual factors, such as anchors [24,25,31].

We note that this study also complements the literature examining 'default' effects on redistribution choices. Defaults are pre-determined choices that will be implemented unless an individual actively changes them [32]. They are related to anchors in that a default option can also act as an anchor. As noted earlier, Dhingra et al (2012) [17] find evidence of what they term a "default pull" on choices in a dictator game with default options. Similar findings are reported in [15 and 23] albeit with respect to cooperation behaviour in a public goods game. Also related is the literature on 'reference points', which people often use to evaluate gains and losses [33], and which have been found to influence bidding behaviour in auctions (e.g. [6]). With regards to impacts on redistribution choices, Charite, Fisman & Kuziemko (2015) [34] find that people's choices are impacted by other people's reference points.

This rest of paper is organised as follows: in the next section we present our research questions and hypotheses. This is followed by the Materials and Methods, after which we present the Results, and finally, the Discussion and Conclusions.

## Identifying anchoring effects

To identify anchoring effects with respect to the initial amount presented to second movers, the order in which the hypothetical first mover transfers were presented to SMs was randomized. Hence, we obtained vectors of responses (SM strategies) for each possible initial amount, or 'anchor' (experimental details are provided in the Experimental Design section). Based on general findings in the literature on anchoring, we hypothesize that SM transfers will be biased towards the anchor (e.g. [11]). We do not aim to identify the precise psychological or cognitive mechanism underlying this anticipated anchoring effect. There are different explanations for anchoring, including 'anchoring-and-adjustment' [1], 'selective accessibility' [35,36] and a close variant of this, 'query theory' [37]. The first of these proposes that individuals use the initial information provided as a starting point (anchor) and reach their final judgment through a process of marginal but insufficient adjustments from this anchor. 'Selective accessibility' and 'query theory' models however suggest that when individuals receive an initial piece of information, they engage in an internal assessment of the validity of this information. Greater weight is placed on the initial information provided, resulting in judgments converging on this initial piece of information.

However, we do not propose to identify whether these (or indeed, other explanations) explain our findings. The main purpose of the present study is, firstly, to assess *whether* the

initial piece of information impacts the redistribution behaviour of individuals in response to social information; and secondly, to identify whether the behavioural strategy that individuals adopt are affected by anchors. The first question has only been addressed by two other studies, as noted in the introduction [16,17]. The second question is novel and has not been addressed previously.

On the one hand, it is possible that all we observe is a magnitude effect–by which subsequent choices are simply adjusted upwards or downwards in response to the initial decision, but no changes in actual strategy occur. Thus, for example, if this were to occur, players classified as 'conformists' would positively condition their choices to the social information provided, albeit with an upward (downward) shift in overall transfers in response to a higher (lower) anchors. Similarly, players classified as 'compensators' and 'unconditional givers' would be expected to continue behaving in line with their type, but with similar upwards (downwards) adjustments. Self-interested contributors however would not be expected to adjust, assuming that they have pure self-interested preferences. On the other hand, it is possible that the reasoning an individual engages in when faced with different anchors affects how they perceive the decision, which could potentially lead to changes in adopted strategy. As noted, this is an exploratory question, and we have no expectations for the pattern of an effect in this regard.

We also consider it possible that the entire order in which FM transfers are presented to SMs may have an effect on choices beyond the effect of the initial amount. To assess possible order effects, we ran a series of tests which are reported in the S1 Appendix. We found no evidence of order effects beyond the impact of the initial amount on SM transfers.

Finally, we acknowledge that there are other contextual factors—such as how the decision is framed—that may influence decisions. Framing effects occur when information is presented in different ways, leading to different interpretations of the context and decision. In our study, it is possible that the first piece of information received (what we term the 'anchor') actually affects choices through a 'framing effect'–i.e. by changing the perception of what the decision context involves. This would be in line with the 'selective accessibility' and 'query theory' models, which propose heavy reliance on the first piece of information to shape one's decision– hence, in this context, the anchoring effect could be akin to a 'framing effect' whereby the frame is provided by the initial information, or 'anchor'.

## Materials and methods

### Experimental design

To explore the influence of first mover (FM) monetary transfers on second mover (SM) redistribution behaviour, we used a 'multi-player dictator game' (MDG) in which SM's could condition their choices on the possible choices of a first mover. At the beginning of the game, participants were randomly assigned to groups of eight players. Within these groups, half of the players were randomly assigned to the role of *allocator* (i.e. 'dictator') and half to the role of *recipient*. Allocators received an endowment of $2 per person; recipients did not receive this endowment.

The next set of instructions informed allocators that that one of them would be randomly selected "by the computer" to make the first transfer and that this amount would be communicated to the other allocators in the group. The instructions specifically read:

"the computer will now <u>randomly</u> select one of you to make a transfer before anyone else. This person will be referred to as the 'first mover'. The transfer made by the first mover will be made visible to all the other participants. Please move to the next page to determine whether you have been selected to be the 'first mover'"

When allocators moved to the next page of the experiment, one of them was informed that s/he had been selected to be the 'first mover'. The FM was then given the option to transfer one of the following amounts from their endowment to the recipients: [$0, $0.10, $0.25, $0.50, $0.75, $1]. These amounts were presented simultaneously on the same page.

Meanwhile, the remaining three allocators moved to another page where they were informed that they had <u>not</u> been selected to be the first mover. These 'second movers' (SMs) were then informed that the FM had been given the choice of transferring one of the six afore-mentioned amounts to the recipients. SMs were then asked to indicate how much they would contribute conditional on each of these possible FM transfers. Fig 1 shows a screenshot of the page that SMs were presented with, outlining these instructions.

Each possible FM transfer was presented to SM's sequentially on separate screens, and in random order–thus implementing our anchoring treatments. SMs indicated their preferred transfers sequentially in response to each of the six possible FM transfers. Hence, we obtained vectors of responses (SM strategies) for each of these six possible anchors. Table 1 shows the sample size for each anchor.

SM transfers were elicited using an open-ended format, such that they could transfer any amount between $0 and $2. Fig 2 shows a sample screenshot of one of these choices offered to SMs.

Each time a SM clicked on "Next" after indicating their preferred transfer, a new screen appeared with another FM transfer. Once the SMs had provided a full vector of responses to each possible FM transfer, the FM's choice was communicated to the SMs.

As a side note, we mention that the strategy method is usually used non-sequentially, i.e. subjects view all possible choices by another subject/other subjects and provide their condi-tional choices simultaneously. Thus, in the standard approach, subjects make their choices under a scenario of "advanced disclosure". Given our interest in identifying whether subjects

---

You have not been randomly-selected selected to be the First Mover.

Please wait a few moments whilst the randomly-selected First Mover makes their transfer.

After the First Mover has made a decision, *you will see how much they have chosen transfer.*

The First Mover has been given the choice of transferring one of the following amounts:

[$0] [$0.10] [$0.25] [$0.50] [$0.75] [$1]

The selected amount will be distributed amongst the four participants who did <u>not</u> receive the $2 bonus, and <u>shared equally among them</u>.

Before seeing what the First Mover has transferred, please indicate what you will contribute in response to each of these possible First Mover transfers.

<u>These transfer decisions are binding</u>. Once you have indicated how much you will transfer *given all possible First Mover transfers*, you will find out how much the First Mover has <u>actually</u> transferred. Then the corresponding amount that you indicated you would transfer in response to this First Mover transfer will be shared equally among the participants who did not receive the $2 bonus.

*Remember: the decisions you make now are binding.*

**Fig 1. Screenshot showing transfer instructions for SMs.**

**Table 1. Summary sample size by anchor.**

| Anchor value (IA) | Sample Size |
|:---:|:---:|
| $0 | 55 |
| $0.10 | 40 |
| $0.25 | 60 |
| $0.50 | 51 |
| $0.75 | 64 |
| $1 | 54 |

would anchor their decisions to the first amount they were presented with, we used a sequential approach. However, to keep our design as close as possible to the standard approach, we opted for advanced disclosure of the FM's choices. Only when choices were to be made, was this done sequentially.

After completing the MDG, allocators (FMs and SMs) were asked to provide an open-ended explanation for their decision–specifically, the question read: "*How did you decide on the amount that you contributed*?" Although this qualitative data lacks the clarity of quantitative measures of social influence on redistribution, it can be used to assess the robustness of the SM classification process. Participants then indicated how much they expected other SMs in their group to contribute on average. Finally, they were asked to provide basic socio-economic information, including their gender, age and income. We expect that redistribution behaviour will be positively influenced by female gender (e.g. [38,39]) and income (due to the income effect).

A custom, web application was used to allow participants to play the game interactively with the other members of their group at the same time. The web application was developed specifically for this experiment primarily using the programming languages PHP, HTML, and Javascript. It was hosted on Amazon EC2 while the experiment was running. This is a fairly novel development in studies using Mechanical Turk subjects (other examples include [40]). Typically, group-based studies using MTurk subjects do not provide interactive platforms for players to play simultaneously with each other. The design in the present study adds realism and urgency to the player's actions, which enhances the validity of group-based decisions.

The experimental instructions can be found in the S1 Data under 'Experimental Instructions'. In addition, a recording of the interactive platform can be found in the following link: http://www.columbia.edu/~ms4403/dictator_game/Dictator%20Game%20Screencast.mp4.

## Analysis procedure

To identify anchoring effects on conditional transfer amounts, firstly, we compare the overall contributions by anchor using a Kruskal-Wallis (KW) test, which is a rank-based nonparametric test used to compare the medians of two or more groups, and is considered the

**Fig 2. Screenshot example—Elicitation of SM transfer in response to FM transfer of $0.**

nonparametric equivalent of the one-way ANOVA. We use the KW test because our examination of residuals (using standardised and quantile normal probability plots) suggest the residuals are not normally distributed; additionally, the Shapiro-Wilk test for normality confirms that the raw data is not normally distributed.

Then, given that we have repeated observations (six) per SM, we assume that observations from the same individual are correlated and hence we opt to use mixed effects regression analyses on the full data set of SM strategy-method transfers, with clustering of standard errors at the individual level. Mixed effects regression is appropriate to model anchoring effects in which SMs are treated as random effects; we do not cluster at the group level as there is no interaction between group members during the strategy data collection stage, so there is no reason that there should be group-level effects. The regression models include dummies for all possible anchors (with IA of $0.50 as the reference) so as to identify the specific impacts of each anchor on transfers and non-linearities. We also run regressions using a dichotomous version of the anchoring variable (where 1 = IA≥$.50 and 0 = IA<$0.50). We tested the assumption that observations by individual SMs are correlated, as required by mixed effects models. Estimation of the intraclass correlation (ICC)–which indicates the correlation among observations within the same 'level' (in this case, the 'individual')–suggests that approximately 85% of the total residual variance in our dependent variable can be accounted for by clustering at the individual level. Wald Chi2 tests and likelihood ratio tests comparing the mixed effects versus linear model confirm that the mixed effects model is suitable for our data.

To identify the impact of anchors on the behavioural strategy that SM's adopt, we first categorize SMs by fitting a linear model (using ordinary least squares) predicting the SM strategy transfer amount by the FM transfer (similar to the approach used in [41,30]). The linear model fitted for each subject was simply:

$$transfer\_amount_i = \beta_0 + \beta_1 FM\_transfer\_amount_i + \epsilon_i$$

$$FM\_transfer\_amount_i \in \{0.0, 0.1, 0.25, 0.50, 0.75, 1.0\}$$

The estimated intercept term, $\beta_0$, and the beta term, $\beta_1$, were used to categorize SMs into four main groups (details can be found in Table 2). To explore whether the adoption of different strategies is affected by the initial information or 'anchor', we conduct a multinomial logistic regression on the different player 'types', as well as a binary logistic regression specifically aimed at addressing whether anchors influence the adoption of a 'self-interested' strategy. Our motivation for focusing on the 'self-interested' type is based on our finding that this particular behavioural strategy appears to be most susceptible to anchors.

As noted earlier, given the focus on this paper on anchoring effects, all results and analyses in this paper pertain solely to SM decisions elicited using the strategy method. Data on FM transfers is not analysed here; however, it is available upon request. The analyses in this paper were conducted using the statistical packages Stata 15 and R.

**Table 2. Classification scheme.**

| Type | Classification | Quantitative criteria |
|------|---------------|----------------------|
| 1 | Conformist | $\beta_1$ significantly positive; y-intercept ($\beta_0$) irrelevant. |
| 2 | Compensator | $\beta_1$ significantly negative; $\beta_0$ irrelevant. |
| 3 | Unconditional giver | $\beta_1$ not significant; $\beta_0$ significantly positive; average transfer>$0.05. |
| 4 | Self-interested | $\beta_1$ not significant; $\beta_0$ not significantly different from zero; average transfer<$0.05 |
| 5 | All other | $R^2$ less than or equal to 0.20 |

## Participants

We used Amazon Mechanical Turk (MTurk) to recruit participants for this experiment. MTurk experiments generally involve low stakes, as participants play from their computers or smartphones, which usually takes less than ten minutes. This allows experimenters to decrease the stakes without compromising the results. This has been confirmed by several studies showing that data collected using MTurk (with low stakes) are of similar quality than those gathered using the standard laboratory [42,43,44].

Consent was obtained at the beginning of the study; participants read a page of text summarising the study and their rights, and if they consented to participate, they could choose to continue or discontinue the study. After providing informed consent, participants were presented with the experimental instructions, followed by two questions testing comprehension. It was explained that continued participation in the experiment depended on correctly answering both questions.

Data was collected from a total of 118 groups of subjects, with eight in each group (four allocators and four recipients). Due to dropouts (n = 39) the final sample consists of 433 allocators (109 FMs, 324 SMs) distributed unevenly among groups. Given the focus on this paper on anchoring effects, all results and analyses in this paper pertain solely to SM decisions elicited using the strategy method. Additionally, as we are interested in individual SM decisions rather than aggregate group decisions, we opt to use the full SM dataset rather than exclude incomplete groups (n = 24 incomplete groups)–we do this because group members did not interact in any way other than by viewing the FM's decision, hence dropouts were not observed by SMs when providing their conditional redistribution choices via the strategy method. For the final pay outs, we always divided the sum of all transfers made among the actual number of recipients in the group, regardless of the number of dropouts. The sample was composed of 43% females; the average age was 33 years and median annual income was $45,000.

This research was approved by Columbia University's Internal Review Board, approval number IRB-AAAM5961.

## Results

### Overview of data

We start by examining the data at the aggregate level, presenting an overview of social information on redistribution decisions. As noted previously, the experiment elicited SM transfers in response to each of six possible FM transfers that were presented sequentially [$0, $0.10, $0.25, $0.5, $0.75, $1]. The distribution of SM contributions in response to each possible FM transfer can be found in S2 Appendix, in addition to a line graph showing mean SM transfers in response to each of these FM transfers.

Overall, mean SM transfers are found to increase modestly with FM transfers. Results of a Friedman test (non-parametric equivalent to a repeated measures ANOVA) suggest that FM transfers have no significant influence on SM transfers overall (Friedman's $\chi^2$ = 8.598, p = 0.1262; Kendall's W (effect size = 0.005). However, additional pairwise paired t-tests and non-parametric Wilcoxon signed-rank tests between mean SM responses (with Bonferroni adjustments to account for multiple testing) suggest that there are *some* significant pairwise differences in SM transfers in response to some FM transfers. For example, there is a significant difference between SM responses to $1 and SM responses to $0, $0.10 and $0.25 (p<0.05 for all tests). Results of these pairwise tests can be found in S3 Appendix.

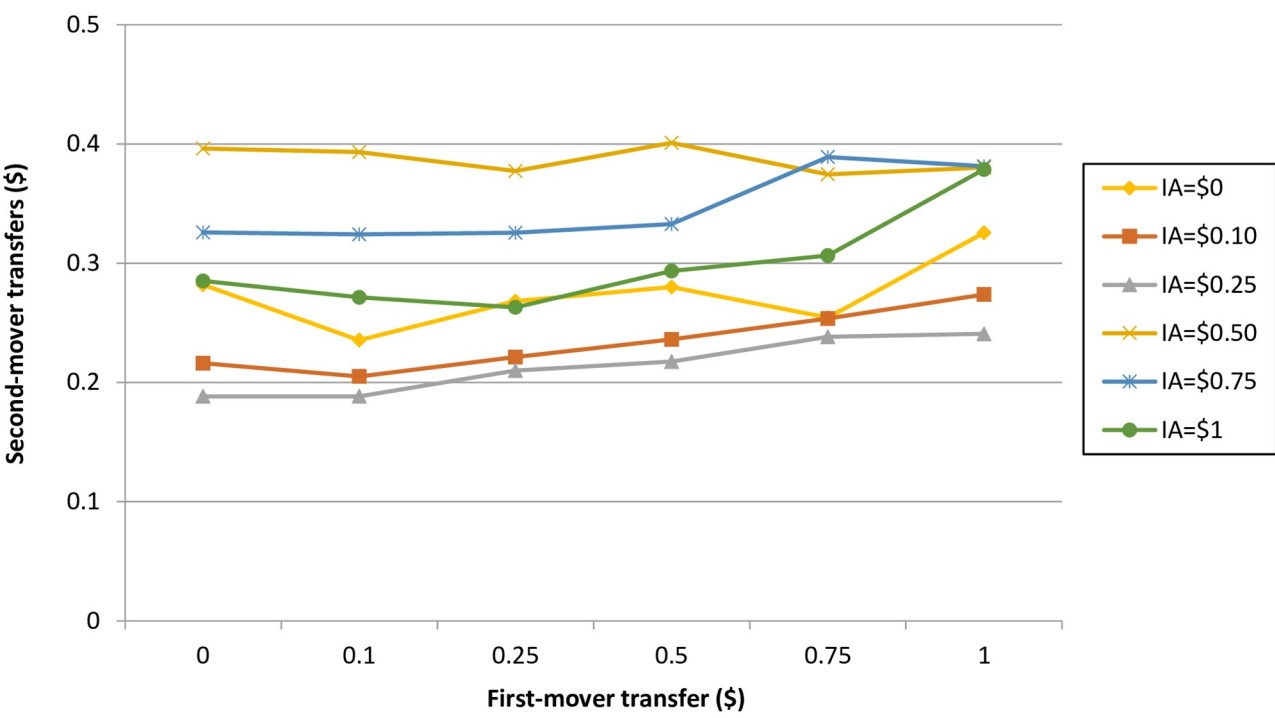

**Fig 3. SM responses to FM transfers disaggregated by initial amount.**

## Anchoring effects on average SM transfers

Fig 3 presents mean SM contributions at each possible FM transfer level disaggregated by IA. Thus, each line in Fig 3 represents mean SM transfers at each FM transfer level for the different anchor treatments; for example, the bottom line with triangular markers shows mean contributions at each possible FM transfer level only for SMs who were presented with an IA of $0.25.

From Fig 3, it appears that mean SM transfers differ by anchor, especially in response to lower FM transfers. For example, SMs in the $0.25 anchor treatment transfer an average of $0.18, while SMs in the $0.75 anchor treatment transfer $0.32. To identify whether there is a statistically significant difference between SM transfers in response to each FM transfer by anchor, we use a Kruskal-Wallis (KW) test, to compare the medians of the anchoring groups. Results indicate that there is a statistically significant difference between SM transfers by anchor treatment in response to FM transfers of $0 (KW: $p = 0.0371$, $\eta^2 = 0.036$)) and $0.10 (KW: $p = 0.0329$, $\eta^2 = 0.036$)). Anchoring treatments have no statistically significant effect on average SM transfers in response to FM transfers of $0.25, $0.50, $0.75 and $1. This shows that anchoring only impacts decisions made in response to selfish FM transfers.

However, it can also be observed that mean SM transfers do not increase linearly with the size of the anchor. For example, the $0.50 anchor appears to elicit the highest mean transfers in response to most FM transfer amounts. Interestingly, contributions in response to the most extreme anchors ($0 and $1) converge in the middle; this suggests that the extreme anchors may lead to more moderate responses. In their review of anchoring studies, Furnham and Boo (2011) [45] report mixed findings regarding the impact of extreme anchors, with some studies finding that extreme anchors generate strong anchoring effects (e.g. [35]) while others find exactly the opposite (e.g. [46]). Our results agree with the latter findings that extreme anchors have weaker anchoring effects. There also appears to be a modest interaction between the IA

and the FM transfer amount, with the IA of $0.50 leading to a fairly flat relationship between IA and FM amount while other IAs suggest positive relationships.

To verify if the differences in average responses to FM transfers by anchor are meaningful, we carry out mixed effects regression analyses on the full data set of individual SM strategy-method transfers. In the models, we include the FM transfer amounts that SMs provided responses to, as well as key socio-economic influences on behaviour (age, gender and income). Additionally, we include a variable representing the order in which FM transfers were presented (first through sixth), to account for possible effects of time or repetition on stated contributions. Studies have shown [47,48] that individuals playing sequential dictator games decrease their contributions round by round, hence we wish to control for this possible source of variation here.

We examine the potential anchoring of transfers to the initial amount (IA) presented to SMs, using dummies for all possible anchors (with IA of $0.50 as the reference) so as to identify specific impacts of each anchor on transfers and non-linearities. We also use a dichotomous version of the anchoring variable (where 1 = IA≥$.50 and 0 = IA<$0.50). This reflects the apparent dichotomised response to the anchors, which we report in the S4 Appendix. Finally, given the apparent interaction between anchor and FM transfer, we also present models with interaction effects. Regression results are presented in Table 3. In S5 Appendix, we report results of similar regressions using only those choices made by SMs in groups without drop-outs, to assess whether there are systematic differences in results when excluding groups with dropouts. As noted previously, dropouts were not observed by SMs when providing their conditional redistribution choices hence there should be no effect of dropouts on choices. Results of these additional regressions confirm that there is no systematic difference in results.

Results in model 1 in Table 3 show that–compared to the reference IA of $0.50 (representing 25% of the endowment)–IAs (anchors) of $0.10 and $0.25 have negative influences on overall transfers, whereas IAs over $0.50 (as well as the IA of $0) do not lead to significantly different SM transfers. In model 2, the dichotomous version of the IA variable has a positive influence on SM transfers, somewhat confirming results in model 1. In addition, results also show that SMs appear to condition their contributions positively to those of FM's. However, the slope is quite modest: for each unit increase in the FM's transfer, SMs increase the amount they transfer by about 5% of the FM's transfer.

Models 3 and 4 include additional terms for interactions between FM transfer and anchors (hence allowing for different slopes). Interaction terms in model 3 show that the slopes associated with anchors of $0.10, $0.25, $0.75 and $1 are positively and significantly different from the slope for the anchor of $0.50 (although this is only weakly significant for the slopes of $0.10, $0.25 and $0.75), partly confirming what can be observed in Fig 3. Ex post tests of the equality of slopes also confirm that all the slopes (except for the slope associated with the anchor of $0.50) are not significantly different to each other. When modelled as dichotomous (model 4), there is no interaction effect. This can be observed visually quite clearly in the figure in the S6 Appendix, which shows SM contributions disaggregated by the dichotomous IA variable. In terms of effect sizes, estimates indicate that the smaller value anchors lead to a reduction of around $0.10-$0.20 in the average amount transferred by SMs (corresponding to about 10–20% of the fair donation amount of $1), depending on the model specification.

Finally, female gender and age positively influence SM transfers, such that older females give more. The positive effect of gender on donations has been found in numerous studies (e.g. [38,39]).

Overall, results indicate that average SM transfers are influenced by the initial FM choice presented to them using the sequential strategy method, thus indicating the presence of an

**Table 3. Regressions on second mover transfers.** The dependent variable is cents transferred per second mover to the recipients. The reference initial amount (IA) level is $0.50.

| | (1) | (2) | (3) | (4) |
|---|---|---|---|---|
| IA = $0 | -11.092 (7.116) | | -14.323* (7.446) | |
| IA = $0.10 | -12.392* (7.498) | | -16.645** (7.904) | |
| IA = $0.25 | -15.420** (6.719) | | -19.021*** (7.077) | |
| IA = $0.75 | -3.122 (7.042) | | -6.852 (7.319) | |
| IA = $1 | -8.165 (7.152) | | -13.214* (7.460) | |
| IA dichotomous (where 1≥$0.50, 0<$0.50) | | 9.284** (3.861) | | 9.944** (4.051) |
| Order in which FM transfer presented | -0.283 (0.200) | -0.283 (0.200) | -0.285 (0.189) | -0.319* (0.189) |
| FM transfer (cents) | 0.051*** (0.015) | 0.051*** (0.015) | -0.026 (0.037) | 0.059*** (0.021) |
| Female | 12.999*** (3.977) | 12.960*** (3.999) | 12.999*** (3.977) | 12.960*** (3.999) |
| Age | 0.477*** (0.170) | 0.458*** (0.171) | 0.477*** (0.170) | 0.458*** (0.171) |
| Income (divided by 1000) | -0.084 (0.054) | -0.094* (0.055) | -0.084 (0.054) | -0.094 (0.055) |
| *Interactions* | | | | |
| IA = $0*FM transfer | | | 0.075 (0.051) | |
| IA = $0.10*FM transfer | | | 0.098* (0.059) | |
| IA = $0.25*FM transfer | | | 0.083* (0.049) | |
| IA = $0.75*FM transfer | | | 0.086* (0.051) | |
| IA = $1*FM transfer | | | 0.117** (0.051) | |
| IA dichotomous*FM transfer | | | | -0.015 (0.030) |
| Constant | 19.257** (8.645) | 7.382 (7.522) | 22.567** (8.787) | 7.166 (7.581) |
| Number of observations | 1884 | 1884 | 1884 | 1884 |
| Number of groups (i.e. SMs) | 313 | 313 | 313 | 313 |
| Wald chi2 | 56.37*** | 55.65*** | 62.28*** | 56.27*** |
| Likelihood ratio test: mixed versus linear model | *** | *** | *** | *** |

Missing data from 10 respondents on income, age and gender (refusal to answer)

Standard errors (clustered at individual level) are shown in parentheses,

* $p < 0.1$,

** $p < 0.05$,

*** $p < 0.01$.

anchoring effect. In addition, the general pattern of SM responses (to the first amount seen *and* in response to all possible FM transfers) suggests a positive relationship between FM contributions and SM contributions (which could be indicative of conformity)—although we do not observe this for the IA of $0.50, for which we observe no relationship between FM and SM

contributions. In the following section, we will examine the extent to which the anchor influences the response strategy selected by individual SM.

## Influence of anchor on individual strategies

SMs were categorized by fitting a linear model (using ordinary least squares) predicting the SM strategy transfer amount by the FM transfer (outlined in the Analysis Procedure section). After fitting a linear model to the data from each participant we categorized them into four main groups, as outlined in Table 2. The classification was guided by theoretical expectations regarding the potential response of individuals to redistribution choices made by others [27]. Briefly, these expectations derive from two broad classes of social preference model; in the first type of model, contributions by others are perceived as complements to one's own contributions due to a desire to conform [49,50]; in the second type of model, contributions by others are seen as substitutes for one's contributions because one mainly cares about recipients' final earnings [51].

Thus, SMs were classed into four main categories: SMs whose transfers are positively correlated with FM transfers are termed 'conformists', whilst those whose transfers are negatively related to those of FM's are termed 'compensators'. We recognize that a positive association between others and one's own contributions may be attributed to other motivations, such as reciprocity, but in this case we are using the definition of conformity as "the act of changing one's behaviour to match the responses of others" ([52, p606]). This definition accounts for any positive conditioning of one's behaviour on the behaviour of others.

In addition, taking into account that SMs may not condition their responses to FM choices, SMs may also be 'self-interested' (zero contribution over all possible FM transfers) or 'unconditional givers' (positive contribution, no relationship with FM transfers).

The distribution of SM types by each of the six anchors can be found in Table 4. A Pearson Chi$^2$ test of the difference in proportions confirms that the proportions of SM types differ significantly between anchors (p = 0.026). This suggests that there are players whose redistribution strategies are susceptible to the anchor. Given the small sub-samples of SM types responding to each anchor, we also present distributions of SM types according to 'low' and 'high' anchors (Fig 4), where 'low' anchors are those IAs that have a value of less than $0.50 and 'high' anchors have a value of $0.50 or more. This figure is intended to complement Table 4 by providing a visual overview of the impact of anchors on the distribution SM types.

Results in Fig 4 clearly show a higher proportion of self-interested players (49.7%) when the IA is low, compared to the proportion of such players (36%) when the IA is 'high'; a two-sample test of proportions indicates that the difference is statistically significant (p = 0.0135). At the same time, the numbers of conformists, unconditional givers and compensators have

**Table 4. Percentage distribution of SM types by anchor.**

| SM Type | Initial Amount (anchor) | | | | | | |
|---|---|---|---|---|---|---|---|
| | **$0** | **$0.10** | **$0.25** | **$0.50** | **$0.75** | **$1** | **Overall** |
| Conformists | 10.91 | 17.50 | 11.67 | 11.76 | 18.75 | 20.37 | 15.12 |
| Compensators | 1.82 | 7.50 | 0 | 15.69 | 3.13 | 3.70 | 4.94 |
| Unconditional givers | 23.64 | 15.00 | 25.00 | 29.41 | 32.81 | 22.22 | 25.31 |
| Self-interested | 47.27 | 55.00 | 48.33 | 31.37 | 39.06 | 37.04 | 42.59 |
| Other | 16.36 | 5.00 | 15 | 11.76 | 6.25 | 16.67 | 12.04 |
| Sample size | 55 | 40 | 60 | 51 | 64 | 54 | 324 |

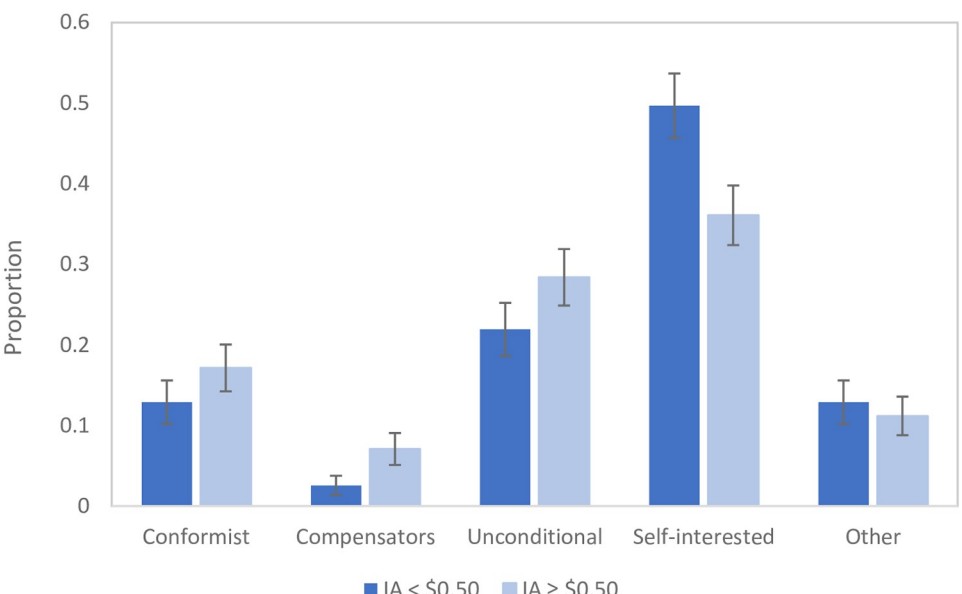

**Fig 4. Distribution of SM types by 'high' or 'low' anchor.**

increased marginally and non-significantly (although a Pearson Chi$^2$ test indicates a weakly significant increase in the proportion of 'compensators' (p = 0.061)).

To assess whether the apparent impact of the IA on the likelihood of adopting self-interested versus all other behavioural strategies can still be observed when controlling for socioeconomic characteristics, we ran a logistic regression on 'self-interested' (where 1 = self-interested player type, and 0 = all other). In Table 5, we report the results of two models, the first using individual dummies representing the different anchoring amounts (with $0,50 as the reference) and the dichotomous version of the IA variable (where 1 = IA≥$0.50, and 0<$0.50). Results of additional multinomial regressions on the individual SM types can be found in S7 Appendix, as complements to the logistic regressions in Table 5. We do not present these results in the main text, as subsample sizes for each player type are below the recommended 10 observations per independent variable [53], leading to potentially biased results. However, results in the multinomial logit models confirm findings in the logistic regression models.

Results in Model 1 in Table 5 show that—when controlling for the socio-economic characteristics of SMs—the higher anchors (above $0.50) significantly reduce the likelihood of adopting a self-interested strategy. Model 2 confirms this to be the case, with lower IAs (namely, $0 and $0.10) having a positive effect on the likelihood of self-interested strategies, compared to the reference of $0.50. This influence of lower anchors appears to be limited to the very lowest values, as there is no relationship between an IA of $0.25 and the likelihood of adopting a self-interested strategy.

With regards to socio-economic characteristics, we observe that females are much less likely to adopt self-interested strategies, compared to all other strategies; this confirms findings that women are more altruistic (e.g. [39]), and adds to the mixed evidence on how conformity relates to gender (e.g. [54,55]).

## Comment: SM expectations

Throughout this paper, we have assumed that SMs either disregard the potential responses of other second movers to FM contributions, or expect non-responsive or conformist behaviour

**Table 5. Logistic regression models of determinants of 'self-interested' SM type.**

| | (1) | (2) |
|---|---|---|
| IA (anchor) presented to SM (where 1 = IA≥$0.50, 0 = IA<0.5) | -0.577** (0.236) | |
| IA = $0 | | 0.709* (0.416) |
| IA = $0.10 | | 1.051** (0.457) |
| IA = $0.25 | | 0.653 (0.413) |
| IA = $0.75 | | 0.340 (0.406) |
| IA = $1 | | 0.208 (0.433) |
| Income (div by 1000) | 0.002 (0.004) | 0.002 (0.003) |
| Age | -0.011 (0.011) | -0.011 (0.011) |
| Female | -0.776*** (0.245) | -0.776*** (0.244) |
| Constant | 0.580 (0.444) | -2.165 (0.499) |
| N | 314 | 314 |
| chi2 | 19.63*** (d.f. = 4) | 321.29*** (d.f = 8) |

Standard errors in parentheses;

* p < 0.1,

** p < 0.05,

*** p < 0.01

[a] Missing data from 10 respondents on income, age and gender (refusal to answer)

of other SMs with respect to FM contributions. However, if the *expected* behaviour of other SMs is negatively correlated with FM contributions, and if SMs mainly condition their responses on their expectations on how other SMs will behave, then this could lead to complications in interpreting SM responses and the classification of redistribution strategies in subsequent sections. However, our analysis of expectations shows that–broadly–SMs consider other SMs to positively condition their contributions to FM contributions. This is true across all SM types. In other words: all SM types expect other SMs to 'conform' to FM contributions, regardless of whether this is the strategy they use or not. We also note that if we control for 'expectations' in the regressions in Table 3, results are unchanged with the exception that expectations are positively and significantly correlated with SM contributions in all models. However, we do not include these models in the main text because the expectations question was not incentivised. As a result, we cannot be sure whether stated expectations influenced contributions, or whether players answered the expectations question in such a way to justify the contributions choices they made in the game. Given this potential problem and the fact that expectations do not affect other variable influences, we opt to omit the expectations variable from the analyses presented in this paper (however, they are available upon request).

## Discussion & conclusions

In this study, we used a multiplayer dictator game to identify how redistribution behaviour is influenced by what others do. Specifically, we examined how second movers (SMs) responded

to contributions by first movers (FMs) to passive recipients, using a strategy game, in which SMs provided a vector of responses to a range of possible FM decisions, ranging from selfish (zero contributions by FM) to a fair split (half of the endowment).

We found that at the aggregate level, SM redistribution choices elicited via a sequential strategy method were positively influenced by the initial amount presented (the anchor). Analysis of SM redistribution choices thus confirm that SMs condition their transfer amounts on the *initial* FM transfer presented to them in the strategy experiment. The size of the effect was found to be small but meaningful. Specifically, smaller value anchors ($0.00, $0.10, and $0.25) were estimated to reduce the amount transferred on average by around $0.10–$0.20 (10–20% of the fair donation amount of $1). While anchoring effects are well-established and have been extensively documented in the empirical literature (see [45] for a review), there is rather less evidence of anchoring effects with regards to monetary transfer decisions. The past literature on anchoring and adjustment has mostly focused on the effect of anchors on judgments, beliefs, and bids for consumer goods, with only a few studies examining how anchors (or related concepts, such as defaults) can affect redistribution or 'fair sharing' behaviour (e.g. [16,17]). Hence, our finding that anchoring effects extend to redistribution decisions is an important contribution to the limited literature. Future studies might explore whether anchors influence other types of pro-social behaviour, such as cooperation.

We also found that the size of the anchor influenced the distribution of behavioural 'types' in our experiment. The impact on the distribution of self-interested individuals appears to be most evident, with higher anchors leading to significantly fewer self-interested players. This adds to the literature showing that the distribution of 'types' may be context-dependent; our focus on how *anchors* in particular influence behavioural strategies is novel and thus a major contribution.

Overall, these findings imply that 'types' may be malleable, and the adoption of a behavioural strategy may be context dependent [24,25]. In particular, we note that self-interested types become less frequent with higher anchors. This suggests there may not just be of one 'type' of self-interested agent. Ubeda (2014) [56] notes that there are two motivations underlying observed self-interested behaviour: on the one hand there is a purely self-interested motivation, in which only one's earnings influence choices, and on the other hand, there are more complex, self-serving motivations, in which there is a tension between pure self-interest and the desire to maintain a positive self-image. An individual of the second type might seek self-justification for selfish behaviour; this justification may be provided in the form of a low IA observed during the initial stages of play. However, if the initial conditions of play involve high anchors, then such a player might struggle to justify a selfish strategy if they also seek to maintain a positive self-image.

Indeed, analysis of open-ended explanations (see S8 Appendix) shows that fewer SMs with self-interested strategies explain their decisions in terms of greed/self-interest under a high anchor (IA $\geq$ $0.50) (52.46% of self-interested subsample), compared to a low anchor (66.23% of self-interested subsample). This difference in proportions however has a small effect size ($h$ = 0.28) and a test of two proportions indicates this is not statistically significant (p = 0.1008). However, these findings can be taken as broadly indicating the possibility that positive self-image is less of a concern among self-interested SMs who received a low anchor.

Further research could examine this apparent switching behaviour among those classed as having self-interested strategies and confirm whether this is only induced by the size of the anchor or whether this occurs in response to other factors. Additionally, it would be valuable to explore in greater detail the cognitive mechanisms underlying self-interested strategies.

We note that Gunnthorsdottir et al. (2007) [57] find that initial cooperative disposition is a good indicator of subsequent behaviour in an experimental setting–in our case we observe

that initial contextual factors may influence an individual's initial disposition as well as the subsequent redistribution strategies of individuals. Thus, not only is individual redistribution behaviour observed to be path dependent, but initial conditions strongly determine the path. If this is indeed the case, it suggests a very fruitful avenue for future research, in which the path dependency of different behaviours in a range of collective decision settings is examined as a function of the initial conditions of play. The outputs from this research may provide critical input into the understanding of how people choose to behave, and the types of citizen that individuals choose to be. It also holds some promise with regards to the potential for self-interested individuals to be 'nudged' towards positive redistribution strategies at critical junctures in time.

## Supporting information

**S1 Appendix. Testing for order effects.**
(DOCX)

**S2 Appendix. SM responses to FM Contributions.**
(DOCX)

**S3 Appendix. Comparing mean SM transfers in response to different FM transfers.**
(DOCX)

**S4 Appendix. SM responses to the initial amount (the anchor).**
(DOCX)

**S5 Appendix. Regressions on second-mover transfers, using only SMs in groups without SM dropouts (n = 279 SMs).** The dependent variable is cents transferred per second mover to the recipients.
(DOCX)

**S6 Appendix. SM responses to FM contributions disaggregated by IA (dichotomous).**
(DOCX)

**S7 Appendix. Multinomial logit model of determinants of SM Type.** Individuals dummies for each anchor (reference category: self-interested).
(DOCX)

**S8 Appendix. Analysing open-ended explanations for transfer decision.**
(DOCX)

**S1 Data. Experimental instructions.**
(DOCX)

## Acknowledgments

Many thanks to Roger Fouquet, Praveen Kujal, Daniele Nosenzo, Natalia Jimenez and Valerio Capraro for valuable comments on this paper. We also wish to thank colleagues at Middlesex Behavioural Economics Group as well as members of the LSE Behavioural Economics group for providing useful feedback on an earlier version of this paper. Finally, we acknowledge the valuable comments and suggestions from two anonymous reviewers.

## Author Contributions

**Conceptualization:** Tanya O'Garra, Matthew R. Sisco.

**Data curation:** Tanya O'Garra, Matthew R. Sisco.

**Formal analysis:** Tanya O'Garra.

**Funding acquisition:** Tanya O'Garra.

**Investigation:** Tanya O'Garra.

**Methodology:** Tanya O'Garra, Matthew R. Sisco.

**Resources:** Tanya O'Garra.

**Software:** Matthew R. Sisco.

**Writing – original draft:** Tanya O'Garra, Matthew R. Sisco.

**Writing – review & editing:** Tanya O'Garra, Matthew R. Sisco.

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
