## [Decision Letter · Decision Letter 0]

21 Oct 2019

PONE-D-19-24674

The Effect of Anchors and Social Information on Behaviour

PLOS ONE

Dear PhD O'Garra,

Thank you for submitting your manuscript to PLOS ONE. After careful consideration, we feel that it has merit but does not fully meet PLOS ONE’s publication criteria as it currently stands. Therefore, we invite you to submit a revised version of the manuscript that addresses the points raised during the review process.

We would appreciate receiving your revised manuscript by Dec 05 2019 11:59PM. To enhance the reproducibility of your results, we recommend that if applicable you deposit your laboratory protocols in protocols.io, where a protocol can be assigned its own identifier (DOI) such that it can be cited independently in the future. For instructions see: http://journals.plos.org/plosone/s/submission-guidelines#loc-laboratory-protocols

We look forward to receiving your revised manuscript.

Kind regards,

Joanna Tyrowicz

Academic Editor

PLOS ONE

**Journal Requirements:**

**Additional Editor Comments (if provided):**

Dear Miss O'Garra,

I have now heard from the referees, and I am happy to invite you to revise your paper and resubmit it for PLOS ONE. While Referee 1 is less critical, Referee 2 raises some important points. I believe that all the raised concerns can be addressed by you in your revised manuscript, so I encourage you to take them in good spirit and attempt to address them.

If you decide to resubmit, please, explain in detail how you addressed the concerns of the referees in a cover letter.

I am looking forward to receiving your submission.

Kind regards,

Joanna Tyrowicz

Reviewers' comments:

Reviewer's Responses to Questions

**Comments to the Author**

1. Is the manuscript technically sound, and do the data support the conclusions?

Reviewer #1: Yes

Reviewer #2: Partly

2. Has the statistical analysis been performed appropriately and rigorously? 

Reviewer #1: Yes

Reviewer #2: Yes

3. Have the authors made all data underlying the findings in their manuscript fully available?

Reviewer #1: Yes

Reviewer #2: No

4. Is the manuscript presented in an intelligible fashion and written in standard English?

Reviewer #1: Yes

Reviewer #2: Yes

5. Review Comments to the Author

Reviewer #1: I think this is a very well written paper, presenting results of an interesting economic experiment on anchoring effects in the context of monetary transfers. I do not have any strong objections, rather only some comments and questions that I hope could be of some inspiration how to improve the paper.

Practical importance of the study

I understand that one of main messages of the paper is that “the choice of behavioural strategies is … affected by normatively irrelevant contextual factors, such as anchors” (p. 4). There is a good discussion of this issue in the paper, but what I am missing is an answer to questions: Why is this outcome important? For what, and in what contexts, can it be useful? I mean providing practical meaning and importance of the study in a broader sense. I see there is a bit of such a discussion in the last paragraph on p. 5. Nevertheless, upon reading the paper, I am left with a feeling that a reader might be not really faced and convinced about practical importance of this study and application of its results. To me, this could be made in the paper more apparent, which would make the contribution of the paper more pronounced.

Focus on anchoring or maybe potential ordering effects?

The authors focus the analysis on the first amount displayed to second-movers in the sequence of considered first-mover transfer amounts. I wonder, however, whether the entire order of questions with different first-mover transfer amounts was not an important factor for decisions of second-movers. For instance, having seen $1 in the first question, a second-mover could differently answer (undertake a different strategy) when faced in the second question with a value of $0 than with a value of $0.75. Furthermore, the behavior could differ depending on whether the sequence was monotonic or not (at least throughout several, e.g., three, initial values). This aspect is paid nearly no attention in the paper, except for including a variable “Order in which FM [first-mover] transfer presented” in models in Table 2. I could not find a definition of levels of this variable in the paper, so I do not understand how this single variable controls for various possible combinations of displaying first-mover transfer amounts. It seems to me that at least in the context of preference elicitation, there is a good part of literature devoted to ordering effects, and this seems to me closely related to the potential role of the order in which the sequence of first-mover amounts is displayed.

Literature references

It seems to me that a little more caution might be needed when describing the current state of the literature. Two specific examples are below.

The authors write: “Most anchoring studies have examined the impact of anchors on … elicited preferences (e.g. Ariely, Loewenstein & Prelec, 2003; Green et al., 1998). Far less research has been conducted on how anchors affect actual behaviour; …” (p. 2)

I wonder about this distinction—in particular, I think that studies concerning elicited preferences can also involve actual behavior. In the context of anchoring, I think that an example of such a study could be Fudenberg et al. (2012). They investigate the role of anchoring when eliciting preferences and their study involve actual payments (probabilistically binding choices). This is just one quick example; I think there are other similarly related studies.

The authors write: “this is the first study to show evidence of anchoring effects influencing monetary transfer decisions” (p. 24). Further in the same paragraph, the author use the phrase “monetary contribution decisions”. I wonder if, for example, the study by Luccasen (2012) does not fit into this description. If yes, that would suggest that the reviewed study was not the first one with respect to the specified research area.

Other questions and comments

$2 seem to be a small amount. Could the result be affected by that? If so, how?

It is clear what possible choices for first movers were (the six discrete values between $0 and $1). What was the set of possible choice for second movers? Was it also a close-ended question or maybe an open-ended one? I have not found explicit information on that in the paper.

“All transfers by FMs and SMs were divided up equally among the recipients in the group.” (p. 8) What happened upon dropouts? Was the entire sum divided by 8 or by a reduced number of recipients?

The question above is related to the dependent variable in models in Table 2. What exactly is the amount—per recipient or total; how does it control dropouts of second-movers or recipients; does it include first-movers’ transfers or only those of second-movers? This clarification could further make clearer the description of the results: for example, “for each unit increase in the FM’s [first-mover’s] transfer, SMs [second-movers] increase the amount they transfer by about 5% of the FM’s transfer” – is the 5% for the entire group of four second-movers? If yes, this would imply an even smaller individual effect, as based on the model: a one-cent increase of first-mover transfer increases the transfer to recipients by 0.05 cent, which would be a 0.05 / 4 transfer increase per second-mover, if I understand the notation correctly.

The last paragraph on p. 18 describes models briefly summarized in Table 3. I think the paragraph could be more explicit, e.g., provide a formula for the models, to make it clear what is meant in Table 3 by beta and y-intercept, and whether any other controls were considered in this modelling.

There is a bit of discussion on a compensating behavioral strategy, on p. 22, among others. I wonder, however, about this referring to the group of compensating second-movers as this is a very small group. If I recovered correctly the numbers of individuals based on Table 4, the number of compensators is 16, out of which 4 are faced with a low anchor (below $0.5) and 12 with a high anchor. This means that some coefficient estimates in the model in Table 5 are based on very few responses, such as 4 (or even less if any of those individuals did not report some of their socio-demographics). I wonder whether it is justified to make any comparisons to such an under-represented group.

“lower anchors also positively (but only weakly) influence the likelihood of adopting conformist and unconditionally giving strategies” (p. 22) Should it not be “higher” instead of “lower” in the first word of this quotation?

Little language corrections might be needed: e.g., “at aggregate level” on p. 11 (an article might be missing); “we’ve” on p. 23 look informal.

References

Fudenberg, D., Levine, D. K., and Maniadis, Z. (2012). On the robustness of anchoring effects in WTP and WTA experiments. American Economic Journal: Microeconomics, 4(2), 131-45.

Luccasen, R. A. (2012). Anchoring effects and participant numbers: Evidence from a public good game. Social Science Quarterly, 93(3), 858-865.

Reviewer #2: On introduction: Overall, the idea of testing the effect of anchoring toward monetary transfer is interesting. The authors take the challenge to bring about a similar concept that has been discussed in different field using different terms (p.2). In this line of thought, the authors statements that this study would be the first one in this path (p. 24 para 2) may need stronger support. This can be done for example by adding more thorough description regarding the conceptual definitions that differentiate anchoring, framing, and social norms. Others may use different terms such a s “default” and “reference points” (Charité, Fisman, & Kuziemko, 2015; Dhingra, Gorn, Kener, & Dana, 2012) for quite similar intention.

In addition, it is perhaps important to also provide reader with a richer understanding toward the meaning of anchor effect utilising this modified dictator game, by explaining how monetary transfer decision used in this study differs with other monetary transfer forms in different context which was also found suffered from anchor effect like in auction (Holst, Hermann, & Musshoff, 2015) and donation (Martin & Randal, 2007). If the difference lies behind the motivation for such behaviour –comparing the use of dictator game (context-free) and charity (context-dependent), then this study could also contribute in the attempt to explain why anchor effect occurred in a context-dependent situation or vice versa (p. 10 para 2 and p. 25 para 1). Further, the authors attempt to explain the behavioural strategies may also refer to studies that have been reported to identify the cognitive mechanism underlie individual redistribution decision (e.g. Crusius, Horen, & Mussweiler, 2012).

On method. The rationale of this study should be put before explaining the experimental method such that we could tell whether the selected procedure has been properly designed and justified. It should also be mentioned if the experiment was originally designed by authors or an adoption/modification from other studies.

The authors could also add more information on how the information on the FM’s transfer being made visible to other players by inserting the screenshot (p.7 para 2). As there are 6 possible FM transfer and 6 possible SM respond, and all those possibilities were presented sequentially to both of them (or it is only randomized to the SM [p. 9, para 2]?), there is actually a chance of order effect (or perhaps, an anchor effect) to the FM too. It is still unclear in this report on how the authors anticipate this issue, e.g. whether the choices were presented in random order etc.

Information regarding the step-by-step instruction during experiment may be better to be written chronologically rather than inserted in the middle (p.8 para 1 and 3) as to make the flow of experiment easier to be understood by a more general reader. Stating the web application used in running this experiment in the text may also considered important when reporting computer based experimental study.

On data report. Descriptive data analysis should be added. Also, the testing for assumption used for the analysis. More importantly, information on goodness of fit of the model for all analysis (e.g. [partial] eta square for ANOVA) should be added, apart from the p-value. Also, the statistical package used in doing the analysis should be named. The authors are advised to see the statistical reporting guidelines for authors provided in the PLOS ONE website for more detail.

6. PLOS authors have the option to publish the peer review history of their article (what does this mean?). If published, this will include your full peer review and any attached files.

Reviewer #1: No

Reviewer #2: No

---

## [Author Response · Author response to Decision Letter 0]

20 Dec 2019

PONE-D-19-24674

The Effect of Anchors and Social Information on Behaviour

PLOS ONE

Comments to the Author

5. Review Comments to the Author

Dear reviewers,

We greatly appreciate the thoughtful comments and suggestions, which we have endeavoured to address to the best of our ability. Any page references are made in relation to the unmarked Manuscript.

Kind regards.

Reviewer #1: I think this is a very well written paper, presenting results of an interesting economic experiment on anchoring effects in the context of monetary transfers. I do not have any strong objections, rather only some comments and questions that I hope could be of some inspiration how to improve the paper.

Practical importance of the study

I understand that one of main messages of the paper is that “the choice of behavioural strategies is … affected by normatively irrelevant contextual factors, such as anchors” (p. 4). There is a good discussion of this issue in the paper, but what I am missing is an answer to questions: Why is this outcome important? For what, and in what contexts, can it be useful? I mean providing practical meaning and importance of the study in a broader sense. I see there is a bit of such a discussion in the last paragraph on p. 5. Nevertheless, upon reading the paper, I am left with a feeling that a reader might be not really faced and convinced about practical importance of this study and application of its results. To me, this could be made in the paper more apparent, which would make the contribution of the paper more pronounced.

Response: We have added the following sentence to the para2 on page 5 highlighting the practical importance of the findings of this study:

Added text:

The practical value of this finding is highly significant, as anchoring effects could potentially be harnessed not only to ‘nudge’ individuals towards single instances of fair sharing, but towards the adoption of more persistent redistributive behaviour.

Focus on anchoring or maybe potential ordering effects?

The authors focus the analysis on the first amount displayed to second-movers in the sequence of considered first-mover transfer amounts. I wonder, however, whether the entire order of questions with different first-mover transfer amounts was not an important factor for decisions of second-movers. For instance, having seen $1 in the first question, a second-mover could differently answer (undertake a different strategy) when faced in the second question with a value of $0 than with a value of $0.75. Furthermore, the behavior could differ depending on whether the sequence was monotonic or not (at least throughout several, e.g., three, initial values). This aspect is paid nearly no attention in the paper, except for including a variable “Order in which FM [first-mover] transfer presented” in models in Table 2. I could not find a definition of levels of this variable in the paper, so I do not understand how this single variable controls for various possible combinations of displaying first-mover transfer amounts. It seems to me that at least in the context of preference elicitation, there is a good part of literature devoted to ordering effects, and this seems to me closely related to the potential role of the order in which the sequence of first-mover amounts is displayed.

Response: We have conducted a series of additional analyses on the data to identify whether there are order effects beyond those related to the impact of the initial amount. These are reported in the Online Appendix S1. We found no evidence of order effects beyond the impact of the initial amount on SM transfers. This is noted on page 12, para2. 

The ’order’ variable in the regressions in Table 2 is an indicator of the order in which FM transfers were presented (first through sixth), as described on page 17, para1. This controls for possible effects of time or repetition on stated contributions but is not intended to identify order effects which are tested as described in Online Appendix S1.

Added text:

We also consider it possible that the entire order in which FM transfers are presented to SMs may have an effect on choices beyond the effect of the initial amount. To assess possible order effects, we ran a series of tests which are reported in the Online Appendix S1. We found no evidence of order effects beyond the impact of the initial amount on SM transfers.

Literature references

It seems to me that a little more caution might be needed when describing the current state of the literature. Two specific examples are below.

The authors write: “Most anchoring studies have examined the impact of anchors on … elicited preferences (e.g. Ariely, Loewenstein & Prelec, 2003; Green et al., 1998). Far less research has been conducted on how anchors affect actual behaviour; …” (p. 2)

I wonder about this distinction—in particular, I think that studies concerning elicited preferences can also involve actual behavior. In the context of anchoring, I think that an example of such a study could be Fudenberg et al. (2012). They investigate the role of anchoring when eliciting preferences and their study involve actual payments (probabilistically binding choices). This is just one quick example; I think there are other similarly related studies.

Many thanks for the suggested reference. We have addressed this missing literature by amending paragraph 2 of the introduction and acknowledging that there are additional studies that examine anchoring effects on valuations with probabilistically binding choices. 

Added references:

Fudenberg, D., Levine, D. K., and Maniadis, Z. (2012). On the robustness of anchoring effects in WTP and WTA experiments. American Economic Journal: Microeconomics, 4(2), 131-45.

Bergman, O., Ellingsen, T., Johannesson, M., & Svensson, C. (2010). Anchoring and cognitive ability. Economics Letters, 107(1), 66-68.

The authors write: “this is the first study to show evidence of anchoring effects influencing monetary transfer decisions” (p. 24). Further in the same paragraph, the author use the phrase “monetary contribution decisions”. I wonder if, for example, the study by Luccasen (2012) does not fit into this description. If yes, that would suggest that the reviewed study was not the first one with respect to the specified research area.

Response: Luccasen (2012) examines anchoring effects on contributions to the public good (reflecting a willingness to cooperate, am interdependent decisions); in contrast, we examine how anchors affect pure redistribution choices (often considered to reflect fairness concerns). 

To address this extra literature, we have amended paragraph 3 in the introduction to introduce a broader (yet limited) literature addressing anchoring effects on pro-social behaviour, of which Luccasen (2012) is one paper. 

We have also amended the wording to be more precise – by referring to “redistribution choices” instead of “monetary contribution decisions”. 

Added references:

Cappelletti, D., Güth, W., & Ploner, M. (2011). Unravelling conditional cooperation. Jena Economic Research Papers, 2011, 047.

Luccasen, R. A. (2012). Anchoring effects and participant numbers: Evidence from a public good game. Social Science Quarterly, 93(3), 858-865

Fosgaard, T. R., & Piovesan, M. (2015). Nudge for (the public) good: how defaults can affect cooperation. PloS One, 10(12), e0145488. 

Dhingra, N., Gorn, Z., Kener, A., & Dana, J. (2012). The default pull: An experimental demonstration of subtle default effects on preferences. Judgment and Decision Making, 7(1), 69.

Other questions and comments

$2 seem to be a small amount. Could the result be affected by that? If so, how?

Response: In fact, $2 is quite standard in Amazon Mechanical Turk experiments, and there have been a number of studies finding that data collected using MTurk (with low stakes) are of similar quality than those gathered using the standard laboratory. We therefore assume that the size of the stake had no significant impact on behaviour. We have added the following text addressing this issue (on page 6):

Added text:

MTurk experiments generally involve low stakes, as participants play from their computers or smartphones, which usually takes less than ten minutes. This allows experimenters to decrease the stakes without compromising the results. This has been confirmed by several studies showing that data collected using MTurk (with low stakes) are of similar quality than those gathered using the standard laboratory (Horton, Rand & Zeckhauser, 2011; Berinski, Huber & Lenz, 2012; Goodman, Cryder & Cheema, 2013; Paolacci & Chandler, 2014). 

Added References:

Goodman, J. K., Cryder, C. E., & Cheema, A. (2013). Data collection in the flat world: The strength and weaknesses of Mechanical Turk samples. Journal of Behavioral Decision Making, 26, 213-224.

Berinski, A. J., Huber, G. A., & Lenz, G. S. (2012). Evaluating online labor markets for experimental research: Amazon.com’s Mechanical Turk. Political Analysis, 20, 351-368.

Paolacci, G., & Chandler, J. (2014). Inside the Turk: Understanding Mechanical Turk as a participant pool. Current Directions Psychological Science, 23, 184-188.

It is clear what possible choices for first movers were (the six discrete values between $0 and $1). What was the set of possible choice for second movers? Was it also a close-ended question or maybe an open-ended one? I have not found explicit information on that in the paper.

Response: Thanks for noting this. Second movers could provide open-ended responses. We have added text on page 9 clarifying this.

Added text:

SM transfers were elicited using an open-ended format, such that they could transfer any amount between $0 and $2

“All transfers by FMs and SMs were divided up equally among the recipients in the group.” (p. 8) What happened upon dropouts? Was the entire sum divided by 8 or by a reduced number of recipients?

Response: For the final pay outs, we always divided the sum of all transfers among the actual number of recipients in the group, regardless of the number of dropouts. This has been clarified on page 8.

The question above is related to the dependent variable in models in Table 2. What exactly is the amount—per recipient or total; how does it control dropouts of second-movers or recipients; does it include first-movers’ transfers or only those of second-movers? This clarification could further make clearer the description of the results: for example, “for each unit increase in the FM’s [first-mover’s] transfer, SMs [second-movers] increase the amount they transfer by about 5% of the FM’s transfer” – is the 5% for the entire group of four second-movers? If yes, this would imply an even smaller individual effect, as based on the model: a one-cent increase of first-mover transfer increases the transfer to recipients by 0.05 cent, which would be a 0.05 / 4 transfer increase per second-mover, if I understand the notation correctly.

Response: The dependent variable in Table 2 is the amount transferred per SM. This has been clarified in main text and in the title of the table. We have also added clarification that the analysis in this paper only uses data from SM responses to the strategy method, given our focus on anchoring effects. This clarification has been added into page 7, para 1, as well as into the new Section 2.3 (Analysis Procedure) on page 16, suggested by reviewer #2 as per the PLOS ONE statistical reporting standards.

Regarding dropouts, we do not consider it relevant to control for dropouts as players never learn whether other players have dropped out when making their strategy decisions - which is what we analyse in this study. Nonetheless, we have re-run new models with only groups which have all SMs, and results are similar; we have reported this in the main text (page 18, para above the table) and added the new regressions in Online Appendix S5. 

As noted above, the results only include the SM decisions, as these were the only ones elicited using the strategy method. FM decisions were elicited via direct response and are not analysed in this study as they do not affect the SM’s responses analysed in this paper. This has been clarified on page 14 (new Section 2.3) with the following text:

Added text:

Given the focus on this paper on anchoring effects, all results and analyses in this paper pertain solely to SM decisions elicited using the strategy method. Data on FM transfers is not analysed here; however, it is available upon request.

The last paragraph on p. 18 describes models briefly summarized in Table 3. I think the paragraph could be more explicit, e.g., provide a formula for the models, to make it clear what is meant in Table 3 by beta and y-intercept, and whether any other controls were considered in this modelling.

Response: This has been done, although we have added in the model in the new added Section 2.3 (Analysis Procedure) which we added in response to a suggestion by Reviewer #2 with regards to the PLOS ONE statistical reporting standards. 

There is a bit of discussion on a compensating behavioral strategy, on p. 22, among others. I wonder, however, about this referring to the group of compensating second-movers as this is a very small group. If I recovered correctly the numbers of individuals based on Table 4, the number of compensators is 16, out of which 4 are faced with a low anchor (below $0.5) and 12 with a high anchor. This means that some coefficient estimates in the model in Table 5 are based on very few responses, such as 4 (or even less if any of those individuals did not report some of their socio-demographics). I wonder whether it is justified to make any comparisons to such an under-represented group.

Response: Thank you very much for spotting this. Indeed, the player ‘type’ sample sizes are too small for multinomial logistic regression. Consequently, we have addressed this issue by replacing the multinomial logit regression with binary logistic regression models in which we examine the determinants of adopting a self-selected strategy versus all other strategies. We have moved the multinomial logit model to the online appendix (S7) for reference with the caveat (stated in the main text), that results may not be reliable due to sample size issues. We have also amended the entire discussion on the results associated with Table 5 (pages 24-26). 

“lower anchors also positively (but only weakly) influence the likelihood of adopting conformist and unconditionally giving strategies” (p. 22) Should it not be “higher” instead of “lower” in the first word of this quotation?

Response: Thanks for noting this. It has been corrected.

Little language corrections might be needed: e.g., “at aggregate level” on p. 11 (an article might be missing); “we’ve” on p. 23 look informal.

Response: These have been corrected.

References

Fudenberg, D., Levine, D. K., and Maniadis, Z. (2012). On the robustness of anchoring effects in WTP and WTA experiments. American Economic Journal: Microeconomics, 4(2), 131-45.

Luccasen, R. A. (2012). Anchoring effects and participant numbers: Evidence from a public good game. Social Science Quarterly, 93(3), 858-865.

Thank you for these references!

Reviewer #2: 

On introduction: Overall, the idea of testing the effect of anchoring toward monetary transfer is interesting. The authors take the challenge to bring about a similar concept that has been discussed in different field using different terms (p.2). In this line of thought, the authors statements that this study would be the first one in this path (p. 24 para 2) may need stronger support. This can be done for example by adding more thorough description regarding the conceptual definitions that differentiate anchoring, framing, and social norms. Others may use different terms such a s “default” and “reference points” (Charité, Fisman, & Kuziemko, 2015; Dhingra, Gorn, Kener, & Dana, 2012) for quite similar intention.

Response: We are grateful for these suggestions, which are indeed relevant to our study (particularly the Dhingra et al, 2012 paper). We have amended para 3 in the introduction to acknowledge this other related literature. We have also added text on page 12-13, acknowledging the relevance of framing effects, which we agree are closely related to anchoring effects.

Added text:

We acknowledge that there are other contextual factors - such as how the decision is framed - that may influence decisions. Framing effects occur when information is presented in different ways, leading to different interpretations of the context and decision. In our study, it is possible that the first piece of information received (what we term the ‘anchor’) actually affects choices through a ‘framing effect’ – i.e. by changing the perception of what the decision context involves. This would be in line with the ‘selective accessibility’ and ‘query theory’ models, which propose heavy reliance on the first piece of information – hence, in this context, the anchoring effect could be akin to a ‘framing effect’.

We have added the following paragraph to page 6 of the Introduction.

Added text:

We note that this study also complements the literature examining ‘default’ effects on redistribution choices. Defaults are pre-determined choices that will be implemented unless an individual actively changes them [32]. They are related to anchors in that a default option can also act as an anchor. As noted earlier, Dhingra et al (2012) [17] find evidence of what they term a “default pull” on choices in a dictator game with default options. Similar findings are reported in [15 and 23] albeit with respect to cooperation behaviour in a public goods game. Also related is the literature on ‘reference points’, which people often use to evaluate gains and losses [33], and which have been found to influence bidding behaviour in auctions (e.g. [6]). With regards to impacts on redistribution choices, Charite, Fisman & Kuziemko (2015) [34] find that people’s choices are impacted by other people’s reference points.

In addition, it is perhaps important to also provide reader with a richer understanding toward the meaning of anchor effect utilising this modified dictator game, by explaining how monetary transfer decision used in this study differs with other monetary transfer forms in different context which was also found suffered from anchor effect like in auction (Holst, Hermann, & Musshoff, 2015) and donation (Martin & Randal, 2007). If the difference lies behind the motivation for such behaviour –comparing the use of dictator game (context-free) and charity (context-dependent), then this study could also contribute in the attempt to explain why anchor effect occurred in a context-dependent situation or vice versa (p. 10 para 2 and p. 25 para 1). 

Response: On page 3, para 2, we have added an explanation of what we mean by ‘redistribution choices’ (added text: “By ‘redistribution’, we refer to decisions to share wealth with others, with no expectation or possibility of benefitting materially from redistribution”). 

This is somewhat different from the effect of anchors on consumer decisions in auctions (as in Holst et al, 2015) which has already been addressed in paragraph 2 of the introduction. We have added this reference with those listed in paragraph 2. As for possible anchoring effects examined in Martin and Randall (2007) we note that the anchoring effect examined in this paper is only considered as a side-issue, and from their findings, it is not possible to disentangle anchoring effects from social information effects. 

Further, the authors attempt to explain the behavioural strategies may also refer to studies that have been reported to identify the cognitive mechanism underlie individual redistribution decision (e.g. Crusius, Horen, & Mussweiler, 2012).

Response: Thank you for this reference, it has been used several times throughout the paper to bolster our argument that context matters (intro, page 3, last sentence; conclusion, page 28, para3) and we have also added a sentence (page 5, end of para 1) acknowledging that different behaviours and strategies may result from very different psychological processes interacting with context, as discussed in the recommended paper. 

Added reference:

Crusius, J., van Horen, F., & Mussweiler, T. (2012). Why process matters: A social cognition perspective on economic behavior. Journal of Economic Psychology, 33(3), 677-685.

On method. The rationale of this study should be put before explaining the experimental method such that we could tell whether the selected procedure has been properly designed and justified. It should also be mentioned if the experiment was originally designed by authors or an adoption/modification from other studies.

Response: The rationale for the study is provided in the Introduction, where we review the literature and identify our research questions and rationale. At the end of para 2, page 4-5 we clarify that to the best of our knowledge, this is the first study to examine anchoring effects using the sequential strategy method. 

The authors could also add more information on how the information on the FM’s transfer being made visible to other players by inserting the screenshot (p.7 para 2). 

Response: We have added two figures: Figure 1 is a screenshot showing how SMs were informed about the transfer choices that FMs had (page 9) and Figure 2 is a screenshot showing how SM transfers were elicited in response to sequential FM transfers (page 9). 

With the addition of Figure 1, there is now some repetition in the paper regarding some of the instructions presented to SMs; to avoid this, we have deleted the repeated text from the main body of the paper (from page 9). 

As there are 6 possible FM transfer and 6 possible SM respond, and all those possibilities were presented sequentially to both of them (or it is only randomized to the SM [p. 9, para 2]?), there is actually a chance of order effect (or perhaps, an anchor effect) to the FM too. It is still unclear in this report on how the authors anticipate this issue, e.g. whether the choices were presented in random order etc.

Response: Thanks for noting this. In fact, FMs were presented the six options simultaneously (now clarified on page 8), so there is no effect of order on their choices. 

We also mention that the SM transfers were elicited using an open-ended format – this has been clarified on page 9 (paragraph between the figures).

Information regarding the step-by-step instruction during experiment may be better to be written chronologically rather than inserted in the middle (p.8 para 1 and 3) as to make the flow of experiment easier to be understood by a more general reader. 

Response: We have rewritten the experimental instructions to make the chronological order clearer.

Stating the web application used in running this experiment in the text may also considered important when reporting computer based experimental study.

Response: The web application was developed specifically for this experiment primarily using the programming languages PHP, HTML, and Javascript. It was hosted on Amazon EC2 while the experiment was running.

We have added this text to page 10.

On data report. Descriptive data analysis should be added. Also, the testing for assumption used for the analysis. More importantly, information on goodness of fit of the model for all analysis (e.g. [partial] eta square for ANOVA) should be added, apart from the p-value. Also, the statistical package used in doing the analysis should be named. The authors are advised to see the statistical reporting guidelines for authors provided in the PLOS ONE website for more detail.

Response: Section 3.1 is a descriptive analysis of the data – it presents an overview of how SMs responded to FM transfers using the strategy method, without any focus on the anchoring effect. A description of the sample characteristics is now found in new section titled ‘Participants’ (page 6-7), which has been added for clarity. 

As suggested by the reviewer, we followed the PLOS ONE statistical reporting guidelines, and added a new section (section 2.3) titled ‘Analysis Procedure’ in the Materials/Methods section. Here we outline the various analyses conducted throughout the paper and we report tests of the assumptions behind the analyses used, such as testing for normality, required for use of ANOVA. We did this using standardised and quantile normality plots and the Shapiro Wilks test. Given that our data (both residuals and raw) are quite non-normal, after careful consideration we decided to remove the ANOVA analyses and only report the non-parametric equivalents (Friedman test for repeated measures and Kruskal Wallis tests comparing contributions by anchor). This is now reported in Section 2.3. To complement the p-values, we have also included effects sizes, as suggested by the reviewer - specifically, eta-square values for Kruskal Wallis tests and Kendall’s coefficient for the Friedman test. For the mixed effects models, we have assessed intraclass correlation and have also added likelihood ratio tests to assess whether linear regression performs better than mixed effects in Table 3. Finally, the statistical packages used in this paper have been specified in the last paragraph in Section 2.3.

---

## [Decision Letter · Decision Letter 1]

8 Jan 2020

PONE-D-19-24674R1

The Effect of Anchors and Social Information on Behaviour

PLOS ONE

Dear PhD O'Garra,

Thank you for submitting your revision to PLOS ONE, as you will see from the referee reports, the paper has been substantially improved in their opinion and I share this judgment. However, there is still some issues, related mostly to presentation and clarity of your writing. Please, take the last effort to perfect your paper (hence: minor revision). Please, mark clearly your improvements relative to the current version of the text upon submission. If possible, please, have the text proof-read by a professional writer, to facilitate the flow and make your article better received by the audience.

We would appreciate receiving your revised manuscript by Feb 22 2020 11:59PM. To enhance the reproducibility of your results, we recommend that if applicable you deposit your laboratory protocols in protocols.io, where a protocol can be assigned its own identifier (DOI) such that it can be cited independently in the future. For instructions see: http://journals.plos.org/plosone/s/submission-guidelines#loc-laboratory-protocols

We look forward to receiving your revised manuscript.

Kind regards,

Joanna Tyrowicz

Academic Editor

PLOS ONE

Reviewers' comments:

Reviewer's Responses to Questions

**Comments to the Author**

1. If the authors have adequately addressed your comments raised in a previous round of review and you feel that this manuscript is now acceptable for publication, you may indicate that here to bypass the “Comments to the Author” section, enter your conflict of interest statement in the “Confidential to Editor” section, and submit your "Accept" recommendation.

Reviewer #1: (No Response)

Reviewer #2: All comments have been addressed

2. Is the manuscript technically sound, and do the data support the conclusions?

Reviewer #1: Yes

Reviewer #2: Partly

3. Has the statistical analysis been performed appropriately and rigorously? 

Reviewer #1: Yes

Reviewer #2: Yes

4. Have the authors made all data underlying the findings in their manuscript fully available?

Reviewer #1: Yes

Reviewer #2: Yes

5. Is the manuscript presented in an intelligible fashion and written in standard English?

Reviewer #1: Yes

Reviewer #2: Yes

6. Review Comments to the Author

Reviewer #1: To me, the paper has been substantially improved upon this revision. I have only minor comments/questions. Other than that, I think the paper is good for publication in the journal.

The authors could make sure that details regarding their study, in particular, in the description of the experiment, are provided sequentially in such an order that is understood to a reader unfamiliar with the study. I mean that a reader learns paragraph-by-paragraph about the study and does not need to wonder what some information means before reaching further parts of the paper. Examples:

(i) p. 7, “… hence drop-outs were not observed by SMs …” – this would not be clear to me as a reader without learning about the experiment structure in next sections of the experiment description. Hence, maybe it would be better if the subsection “Participants” was placed at the end of the section describing methods.

(ii) p. 14, the paragraph below the equations says about the analysis focused only on self-interested strategy. At the stage there, it is not clear why it is done so, and it sounds confusing.

The paragraph on p. 11 describes potential explanations. I know that it is a side discussion for this paper, nevertheless, the paragraph sounds a little misleading to me. It suggests as if there are only three possible explanations, although I think there could be many more, and these other explanations are provided in the literature too. For example, learning, fatigue or willingness to behave consistently. I do not mean to largely extend this side discussion in the paper but just to be careful in order not to suggest fewer explanations than the literature actually provides.

Getting to known the screen as displayed to second-movers (Figure 1), it seems to me that the paper could include a short note on advanced disclosure. There was a reason for which the authors decided to present upfront all possible first-movers’ amounts to second-movers, and this could be explained. Especially that in some literature, advanced disclosure has been suggested to mitigate order/anchoring effects – e.g., Day, B., Bateman, I. J., Carson, R. T., Dupont, D., Louviere, J. J., Morimoto, S., Scarpa, R., and Wang, P. (2012). Ordering effects and choice set awareness in repeat-response stated preference studies. Journal of Environmental Economics and Management, 63(1), 73-91.

The discussion on p. 26 talks about the analysis of expectations. I do not know whether it is my mistake but I have missed information in the paper how the data on expectations was collected (e.g., what type of questions and in which part of the experiment). As this is used for the discussion in the paper, I think it would be good to inform a reader about these few details.

Given the equation on p. 14, I think the text in Table 3 could be adjusted. Specifically, the text informs about “beta”, while the equation includes two betas.

In Table 4, the word “selfish” could probably be changed for “self-interested” for consistency.

The text refers to section numbers, although the numbers are not present in the paper.

There are some punctuation and other mistakes that need corrections. Examples:

- p. 10, “(other examples include [40].” – missing end bracket

- p. 11, “[41,42]and” – missing space

- p. 13, “a Kruskal-Wallis (KW) test, which a rank-based nonparametric test” – missing “is”

- p. 17, “transfers. in the models” – a large letter needed

- p. 24, “(with $0,50 as the reference)” – a dot instead of a comma

- inconsistent use of the word “dropouts” either with a dash or without it

Reviewer #2: This version has been highly improved; the richness of the data is presented clearly. I have only several comments.

1. On flow of writing: It might be helpful if the author mentioned clearly in the beginning that this study (particularly for RQ 2) is an exploratory and therefore readers could expect many interesting findings from this study following their hypotheses. As for the RQ 1, I suppose that actually the authors proposed an explicit hypothesis but it was written a bit far behind (p. 11 par 1), and some additional hypotheses about the anchor effect on female and age in the previous part (p. 10, par 1). I wonder if the “Identifying Anchor Effect” part (p. 10) can be moved up to right after Introduction. This is to ensure the link between literatures, research questions and hypothesis proposed by the author kept closed and therefore easily understood by the reader.

2. I understand that the authors have no intention to go further to explain the cognitive mechanism behind redistributive behaviour in this setting. But, since it is also written in the beginning that the focus of this study is redistributive behaviour, rather than cooperative behaviour (p.3 par 2) it may also important to relate this argument here with the argument of explanation used in the Result and Discussion part, that is: the influence of social information toward redistribution and behavioural strategy.

3. On reporting data and discussion. The authors may consider to add explanation of the result based on the effect size. The effect size will be important to be used when discussion the result and what are suggested further research, regardless the p-value. I really appreciate the qualitative data on types of behavioural strategy. I agree this is very important findings. Just one little questions regarding the open ended data processing (whether you are using interrater or not, etc.). This is perhaps can be added, just to be sure.

7. PLOS authors have the option to publish the peer review history of their article (what does this mean?). If published, this will include your full peer review and any attached files.

Reviewer #1: No

Reviewer #2: No

---

## [Author Response · Author response to Decision Letter 1]

5 Feb 2020

(Note: these responses have also been uploaded as a Word document)

Reviewer #1: To me, the paper has been substantially improved upon this revision. I have only minor comments/questions. Other than that, I think the paper is good for publication in the journal.

The authors could make sure that details regarding their study, in particular, in the description of the experiment, are provided sequentially in such an order that is understood to a reader unfamiliar with the study. I mean that a reader learns paragraph-by-paragraph about the study and does not need to wonder what some information means before reaching further parts of the paper. Examples:

(i) p. 7, “… hence drop-outs were not observed by SMs …” – this would not be clear to me as a reader without learning about the experiment structure in next sections of the experiment description. Hence, maybe it would be better if the subsection “Participants” was placed at the end of the section describing methods.

As suggested, the ’Participants’ section has been moved to the end of the methods section. Given this change of location, some of the text in this section has been adjusted or moved within the section to ensure that the text flows correctly, hence improving clarity of reading. In addition, the introductory sentences of the ‘Experimental Design’ section have also been adjusted due to the change in location, to improve readability and flow. 

(ii) p. 14, the paragraph below the equations says about the analysis focused only on self-interested strategy. At the stage there, it is not clear why it is done so, and it sounds confusing.

To make this clearer, we have adjusted the text on page 14 to read:

“To explore whether the adoption of different strategies is affected by the initial information or ‘anchor’, we conduct a multinomial logistic regression on the different player ‘types’, as well as a binary logistic regression specifically aimed at addressing whether anchors the adoption of a ‘self-interested’ strategy. Our motivation for focusing on the ‘self-interested’ type is based on our finding that this particular behavioural strategy appears to be most susceptible to anchors.”

We have also removed the sentence saying that the multinomial regression has been placed in the Online Appendix, and instead make this point – and the justification for doing so – in the Results section. This makes for more clarity in reading the Analysis Procedures section.

The paragraph on p. 11 describes potential explanations. I know that it is a side discussion for this paper, nevertheless, the paragraph sounds a little misleading to me. It suggests as if there are only three possible explanations, although I think there could be many more, and these other explanations are provided in the literature too. For example, learning, fatigue or willingness to behave consistently. I do not mean to largely extend this side discussion in the paper but just to be careful in order not to suggest fewer explanations than the literature actually provides.

The proposed psychological mechanisms presented on page 6-7 are the main ones used to explain anchoring in the literature (reviewed in Cho, 2011), which is why we refer to these. However, we have acknowledged the possibility of other explanations as follows on page 7:

“However, we do not propose to identify whether these (or indeed, other explanations) explain our findings.”

Getting to known the screen as displayed to second-movers (Figure 1), it seems to me that the paper could include a short note on advanced disclosure. There was a reason for which the authors decided to present upfront all possible first-movers’ amounts to second-movers, and this could be explained. Especially that in some literature, advanced disclosure has been suggested to mitigate order/anchoring effects – e.g., Day, B., Bateman, I. J., Carson, R. T., Dupont, D., Louviere, J. J., Morimoto, S., Scarpa, R., and Wang, P. (2012). Ordering effects and choice set awareness in repeat-response stated preference studies. Journal of Environmental Economics and Management, 63(1), 73-91.

We have added the following paragraph to the Experimental Design section (page 11) clarifying our use of “advanced disclosure” of FM transfers to SMs:

“As a side note, we mention that the strategy method is usually used non-sequentially, i.e. subjects view all possible choices by another subject/other subjects and provide their conditional choices simultaneously. Thus, in the standard approach, subjects make their choices under a scenario of “advanced disclosure”. Given our interest in identifying whether subjects would anchor their decisions to the first amount they were presented with, we used a sequential approach. However, to keep our design as close as possible to the standard approach, we opted for advanced disclosure of the FM’s choices. Only when choices were to be made, was this done sequentially.”

The discussion on p. 26 talks about the analysis of expectations. I do not know whether it is my mistake but I have missed information in the paper how the data on expectations was collected (e.g., what type of questions and in which part of the experiment). As this is used for the discussion in the paper, I think it would be good to inform a reader about these few details.

This has been clarified in the Experimental Design section (page 11) as follows:

“Participants then indicated how much they expected other SMs in their group to contribute on average.”

Given the equation on p. 14, I think the text in Table 3 could be adjusted. Specifically, the text informs about “beta”, while the equation includes two betas. 

Thank you for noting this. Table 3 has been adjusted to reflect the two different betas used. 

In Table 4, the word “selfish” could probably be changed for “self-interested” for consistency. This has been fixed

The text refers to section numbers, although the numbers are not present in the paper.

These have been removed, and/or section titles referred to instead of section numbers.

There are some punctuation and other mistakes that need corrections. Examples:

- p. 10, “(other examples include [40].” – missing end bracket fixed

- p. 11, “[41,42]and” – missing space fixed

- p. 13, “a Kruskal-Wallis (KW) test, which a rank-based nonparametric test” – missing “is” fixed

- p. 17, “transfers. in the models” – a large letter needed fixed

- p. 24, “(with $0,50 as the reference)” – a dot instead of a comma fixed

- inconsistent use of the word “dropouts” either with a dash or without it thanks for noting this – we have chosen to go with “dropout” and have corrected other spellings

Reviewer #2: This version has been highly improved; the richness of the data is presented clearly. I have only several comments.

1. On flow of writing: It might be helpful if the author mentioned clearly in the beginning that this study (particularly for RQ 2) is an exploratory and therefore readers could expect many interesting findings from this study following their hypotheses. As for the RQ 1, I suppose that actually the authors proposed an explicit hypothesis but it was written a bit far behind (p. 11 par 1), and some additional hypotheses about the anchor effect on female and age in the previous part (p. 10, par 1). I wonder if the “Identifying Anchor Effect” part (p. 10) can be moved up to right after Introduction. This is to ensure the link between literatures, research questions and hypothesis proposed by the author kept closed and therefore easily understood by the reader.

On page 4 (end of para 1) we have added the following text clarifying that this is exploratory research:

“This is exploratory research, and as such, we have no expectations about the size or direction of anchoring effects on the distribution of ‘types’ in the population under study. Our aim is mainly to identify whether the choice of behavioural strategy is affected by normatively irrelevant contextual factors, such as anchors.”

We also moved the section ‘Identifying Anchoring Effects’ so that is now right after the Introduction, as suggested. Given this change of location, some of the text in this section has been adjusted or moved within the section to ensure flow and readability. Table 1 and the accompanying text (which were in the ‘Identifying Anchoring Effects’ section) have now been moved to the ‘Experimental Design’ section, with some minor adjustments to the accompanying text. This has been done to maintain readability, given the change of location of various sections in the paper.

2. I understand that the authors have no intention to go further to explain the cognitive mechanism behind redistributive behaviour in this setting. But, since it is also written in the beginning that the focus of this study is redistributive behaviour, rather than cooperative behaviour (p.3 par 2) it may also important to relate this argument here with the argument of explanation used in the Result and Discussion part, that is: the influence of social information toward redistribution and behavioural strategy.

We have acknowledged that anchors may affect other pro-social behaviours, by adding a sentence in the Discussion and Conclusions (page 29) proposing that future research might address how anchors affect other pro-social behaviours, such as cooperation. 

3. On reporting data and discussion. The authors may consider to add explanation of the result based on the effect size. The effect size will be important to be used when discussion the result and what are suggested further research, regardless the p-value. I really appreciate the qualitative data on types of behavioural strategy. I agree this is very important findings. Just one little questions regarding the open ended data processing (whether you are using interrater or not, etc.). This is perhaps can be added, just to be sure.

Effect sizes with respect to results in Table 2 have now been reported on page 21, and effect sizes are now mentioned in the Discussion and Conclusions section (page28), in addition to the statistical significance, as suggested.

As for the open-ended data, it is explained in the Online Appendix S8 that the data was coded by three people and final codes agreed on through discussion. We have also added a measure of interrater reliability (Gwet’s AC, value of 0.56) to the Online Appendix.

---

## [Editor Report · Decision Letter 2]

11 Feb 2020

PONE-D-19-24674R2

The Effect of Anchors and Social Information on Behaviour

PLOS ONE

Dear PhD O'Garra,

thank you for submitting the revised version of your study. I think both referees would be fully satisfied. I read your paper carefully and there are two issues which I find a bit problematic. Feel free to answer me directly by emailing to j.tyrowicz@uw.edu.pl if you find these questions simply wrong. However, if I am right, I would expect you to fix those minor issues, upon which your paper can be accepted at PLOS ONE.

Issue #1. When I look at Table 2, it is clear to me what is the reference level for your dummies in Column 1, but in Column 3 you interact your IA dummies with the amount. In principle it can be informative, but I am confused about the base levels in this case. As you have 0 for IA!=1 and a given amount, it seems to me that the base level in Column 3 is both 0 for each level of IA and on top of that IA=0.5$. I find that confusing and given the interaction term, you can use all the levels in Column 3. 

Issue #2. Table 2 reports a mixed effects regression, but nowhere in your paper do I find the assumption concerning the standard error, i.e. I think you should cluster standard errors at SM level, because presumably all the responses of a given SM are driven by the same decision rule (i.e. they are not independent). If Table 2 reports clustered standard errors, I think it should be made salient. If they are not clustered, I fear that you may need to re-estimate the model and report the new standard errors in your Table 2.

These are minor issues. If re-estimating your model necessitates other changes in text, please do so. As minor minor points, I can bring to your attention the following:

* I think it is typical to use present simple tense in reporting the findings of the literature and your own (rather than present continuous, or other continuous). 

* Perhaps your paper is a bit long both in words and in the illustrations. I leave it at your discretion for the text. As to the illustrations, personally, I see little value to Figure 1 and 2, they can very well be reported in one Table (if need be) and it may confuse some readers not to see the CI whiskers (even despite your notes). 

Please note that the above comments have no influence on whether or not your paper is accepted for publication in PLOS ONE. The decision is positive and will not change if your results change (e.g. because of clustering). These are my requests made in the interest of presenting the highest quality research to our audience. Note also that if I am mistaken in the two issues numbered above, I am happy to hear your comments. Maybe they help us clarify the text so that other readers would not be confused.

We would appreciate receiving your revised manuscript by Mar 27 2020 11:59PM. To enhance the reproducibility of your results, we recommend that if applicable you deposit your laboratory protocols in protocols.io, where a protocol can be assigned its own identifier (DOI) such that it can be cited independently in the future. For instructions see: http://journals.plos.org/plosone/s/submission-guidelines#loc-laboratory-protocols

A letter that responds to my points raised above. This letter should be uploaded as separate file and labeled 'Response to Reviewers'.A marked-up copy of your manuscript that highlights changes made to the original version. This file should be uploaded as separate file and labeled 'Revised Manuscript with Track Changes'.An unmarked version of your revised paper without tracked changes. This file should be uploaded as separate file and labeled 'Manuscript'.

Please note while forming your response, that you may have the opportunity to make the peer review history publicly available. The record will include editor decision letters (with reviews) and your responses to reviewer comments. If eligible, we will contact you to opt in or out.

We look forward to receiving your revised manuscript.

Kind regards,

Joanna Tyrowicz

Academic Editor

PLOS ONE

---

## [Author Response · Author response to Decision Letter 2]

25 Feb 2020

Response to suggestions by editor

Issue #1. When I look at Table 2, it is clear to me what is the reference level for your dummies in Column 1, but in Column 3 you interact your IA dummies with the amount. In principle it can be informative, but I am confused about the base levels in this case. As you have 0 for IA!=1 and a given amount, it seems to me that the base level in Column 3 is both 0 for each level of IA and on top of that IA=0.5$. I find that confusing and given the interaction term, you can use all the levels in Column 3. 

Response: following our email conversation, I have added a sentence to the table clarifying that the reference level for the IA is $0.50 (this text is also included in the main text).

Issue #2. Table 2 reports a mixed effects regression, but nowhere in your paper do I find the assumption concerning the standard error, i.e. I think you should cluster standard errors at SM level, because presumably all the responses of a given SM are driven by the same decision rule (i.e. they are not independent). If Table 2 reports clustered standard errors, I think it should be made salient. If they are not clustered, I fear that you may need to re-estimate the model and report the new standard errors in your Table 2. These are minor issues. If re-estimating your model necessitates other changes in text, please do so. 

Response: as suggested, I have re-run the models with clustering of standard errors (using vce(cluster id) as suggested). This has resulted in some changes to the regression results which have now been updated in the manuscript. As a result, I have had to rewrite some of the text to account for these changes. For consistency, I have also re-run the extra models reported in Online Appendix S5 also with clustering of standard errors at the individual (MM) level. 

As minor minor points, I can bring to your attention the following:

* I think it is typical to use present simple tense in reporting the findings of the literature and your own (rather than present continuous, or other continuous). 

Thanks for noting this, it has been corrected.

* Perhaps your paper is a bit long both in words and in the illustrations. I leave it at your discretion for the text. As to the illustrations, personally, I see little value to Figure 1 and 2, they can very well be reported in one Table (if need be) and it may confuse some readers not to see the CI whiskers (even despite your notes). 

Response: (note: we clarified by email that you were referring actually to Figs 3 and 4). To shorten the length of the paper, I have put Figure 3 in the Online Appendix S2 together with the distributions of SM responses to FM transfers. I have not included standard error bars as this is not correct with repeated measures data (as clarified in Estes, 1997) and box plots are not suitable for continuous two-way data. However, by placing the line graph together with the histograms showing the distributions of responses to each FM transfer, it should be clear to the reader that these averages come from distributions of responses. To further emphasise the link between the distributions and the line graph, I have added the following sentence to the Online Appendix:

“In the following figure we summarise the above distributions in the form of a line graph depicting mean SM transfers in response to each FM transfer.”

---

## [Editor Report · Decision Letter 3]

19 Mar 2020

The Effect of Anchors and Social Information on Behaviour

PONE-D-19-24674R3

Dear Dr. O'Garra,

We are pleased to inform you that your manuscript has been judged scientifically suitable for publication and will be formally accepted for publication once it complies with all outstanding technical requirements.

With kind regards,

Joanna Tyrowicz

Academic Editor

PLOS ONE
---

## [Editor Report · Acceptance letter]

25 Mar 2020

PONE-D-19-24674R3 

The Effect of Anchors and Social Information on Behaviour 

Dear Dr. O'Garra:

I am pleased to inform you that your manuscript has been deemed suitable for publication in PLOS ONE. Congratulations! Your manuscript is now with our production department. 

With kind regards,

on behalf of

Professor Joanna Tyrowicz 

Academic Editor

PLOS ONE